# Look-Back and Look-Forward Durations and the Apparent Appropriateness of Ambulatory Antibiotic Prescribing

**DOI:** 10.3390/antibiotics11111554

**Published:** 2022-11-04

**Authors:** Adriana Guzman, Tiffany Brown, Ji Young Lee, Michael A. Fischer, Mark W. Friedberg, Kao-Ping Chua, Jeffrey A. Linder

**Affiliations:** 1Department of Medicine, Division of General Internal Medicine, Northwestern University Feinberg School of Medicine, Chicago, IL 60202, USA; 2Department of Medicine, Section of General Internal Medicine, Boston Medical Center, Boston University Chobanian & Avedisian School of Medicine, Boston, MA 02118, USA; 3Blue Cross Blue Shield of Massachusetts, Boston, MA 02199, USA; 4Department of Medicine, Division of General Internal Medicine, Brigham and Women’s Hospital, Boston, MA 02115, USA; 5Susan B. Meister Child Health and Evaluation Research Center, Department of Pediatrics, University of Michigan Medical School, Ann Arbor, MI 48109, USA

**Keywords:** antimicrobial stewardship, cohort studies, anti-bacterial agents, drug utilization, quality of healthcare

## Abstract

Ambulatory antibiotic stewards, researchers, and performance measurement programs choose different durations to associate diagnoses with antibiotic prescriptions. We assessed how the apparent appropriateness of antibiotic prescribing changes when using different look-back and look-forward periods. Examining durations of 0 days (same-day), −3 days, −7 days, −30 days, ±3 days, ±7 days, and ±30 days, we classified all ambulatory antibiotic prescriptions in the electronic health record of an integrated health care system from 2016 to 2019 (714,057 prescriptions to 348,739 patients by 2391 clinicians) as chronic, appropriate, potentially appropriate, inappropriate, or not associated with any diagnosis. Overall, 16% percent of all prescriptions were classified as chronic infection related. Using only same-day diagnoses, appropriate, potentially appropriate, inappropriate, and not-associated antibiotics, accounted for 14%, 36%, 22%, and 11% of prescriptions, respectively. As the duration of association increased, the proportion of appropriate antibiotics stayed the same (range, 14% to 18%), potentially appropriate antibiotics increased (e.g., 43% for −30 days), inappropriate stayed the same (range, 22% to 24%), and not-associated antibiotics decreased (e.g., 2% for −30 days). Using the longest look-back-and-forward duration (±30 days), appropriate, potentially appropriate, inappropriate, and not-associated antibiotics, accounted for 18%, 44%, 20%, and 2% of prescriptions, respectively. Ambulatory programs and studies focused on appropriate or inappropriate antibiotic prescribing can reasonably use a short duration of association between an antibiotic prescription and diagnosis codes. Programs and studies focused on potentially appropriate antibiotic prescribing might consider examining longer durations.

## 1. Introduction

Owing to their size and clinical detail, large databases—including electronic health record and claims data—are commonly used to evaluate the appropriateness of ambulatory antibiotic prescribing. However, the indication for antibiotic prescriptions large databases is not always clear. Consequently, antibiotic stewards, researchers, and performance measurement programs using large data sets must choose how to associate prescriptions with diagnoses in a fair and clinically reasonable manner.

Two choices include over what duration to associate antibiotic prescriptions with diagnosis codes and whether to look “backwards” or look “forwards” in time. The simplest approach might be to only consider diagnoses codes occurring on the same date as the antibiotic prescription. However, a patient could call and receive an antibiotic prescription during a telephone encounter with no same-day diagnosis, then make an in-person visit several days later with an antibiotic-appropriate diagnosis. In this situation, considering diagnosis codes during a look-forward period would allow the classification of antibiotic appropriateness. Alternatively, a patient could be seen in person for a potentially antibiotic-appropriate diagnosis, not receive a same-day antibiotic, send a follow-up electronic message days later, and receive an antibiotic. In this situation, considering diagnosis codes during a look-back period would similarly allow classification of antibiotic appropriateness. 

Owing in part to the lack of prior literature on the impact of these choices on assessments of antibiotic appropriateness, prior antibiotic prescribing studies and performance measurement programs have used widely varying look-back and look back-and-forward durations, such as 1 day [1], 2 days [2,3,4], 3 days [5,6,7,8,9,10], 5 days [11], 7 days [12,13,14], and 30 days [15]. The impact of these choices on the seeming appropriateness of antibiotic prescribing is unclear and, to our knowledge, has not been previously examined. In this retrospective cohort study of ambulatory antibiotic prescribing in an integrated academic health system over four years, we assessed how the apparent appropriateness of antibiotic prescribing changed when using different look-back and look-forward durations. Results of this study could inform the best methodological practices for assessing antibiotic appropriateness in EHR data.

## 2. Materials and Methods

### 2.1. Study Design and Setting

We conducted a retrospective cohort study to assess the appropriateness of all ambulatory, oral antibacterial antimicrobial prescriptions ordered in the EHR between 1 January 2016 to and 31 December 2019 by clinicians at Northwestern Medicine. Northwestern Medicine is a large, integrated academic health delivery system with hundreds of sites, about 4000 physicians, and 30,000 staff in the Chicago area (IL, USA) [16]. Northwestern uses a single EHR (Epic; Verona, WI, USA) in which work is organized into “encounters.” The Northwestern University Institutional Review Board approved this study with a waiver of informed consent for retrospective review of EHR data. 

### 2.2. Antibiotics

We used an EHR-based medication grouper to identify all outpatient oral, antibacterial antimicrobial prescriptions. We excluded non-oral antibiotics and antibiotics most often used for urinary tract infection prophylaxis (e.g., methenamine). The unit of analysis was the prescription. We included antibiotic prescriptions regardless of encounter type (e.g., in-person, telephone, patient portal, refills, and others). 

### 2.3. Associating Antibiotics with Diagnoses

Clinicians could order antibiotics with zero, one, or more than one diagnosis codes in the EHR. Diagnoses were not required for all encounter types. For example, at least one diagnosis was required for an in-person encounter, but not for telephone or patient portal encounters. Although antibiotics are often prescribed at the time of an encounter, clinicians have the option of opening a previously started encounter on subsequent days and prescribing antibiotics. Similarly, diagnoses are usually assigned at the time of an encounter but sometimes are assigned later. Clinicians can also assign diagnoses and prescribe related antibiotics in different encounters (e.g., an in-person visit with a diagnosis and a subsequent telephone encounter with an antibiotic prescription). In the EHR, there is no requirement that the same clinician prescribe the antibiotic and choose the diagnosis within or between encounters. At the time of the study, before the COVID-19 pandemic, Northwestern Medicine did not have synchronous video visits.

### 2.4. Antibiotic Appropriateness

Using diagnosis codes, we categorized all antibiotic prescriptions into five mutually exclusive groups: chronically used antibiotics (e.g., for acne, chronic sinusitis), associated with an antibiotic-appropriate diagnosis (e.g., urinary tract infection, streptococcal pharyngitis, bacterial pneumonia), associated with a potentially antibiotic-appropriate diagnosis (e.g., acute sinusitis, acute suppurative otitis media, acute pharyngitis), associated with inappropriate diagnosis (e.g., acute bronchitis, acute upper respiratory infection, cough), or not associated with any diagnosis. 

To identify chronically used antibiotics, we examined diagnosis codes and problem list diagnoses present in the 6 months prior to antibiotic prescribing for 690 International Classification of Diseases 10, Clinical Modification (ICD-10-CM) codes that could be associated with a chronic infection or might be indications for chronic antibiotic use (e.g., chronic osteomyelitis, acne, cystic fibrosis, emphysema) [12,17]. If any of these 690 codes were present during this period, we considered the antibiotic to be for chronic antibiotic use, regardless of any other associated diagnosis codes during this look back period. 

To assign the remaining antibiotics to one of the four other appropriateness categories, we classified all diagnosis codes within a defined duration (described below) as always, sometimes, or never justifying antibiotic prescribing using a previously developed scheme of all 94,249 ICD-10-CM diagnosis codes [5]. If at least one of the 9495 “always” diagnosis codes were present, we classified the prescription as “antibiotic-appropriate.” If at least one of the 11,143 “sometimes” diagnosis codes but no “always” code was present, we classified the prescription as “potentially antibiotic-appropriate.” If at least one “never” code but no “always” or “sometimes” code was present, we classified the antibiotic prescription as “inappropriate.” If there were no diagnosis codes during the defined period, we considered the prescription not to be associated with any diagnosis.

### 2.5. Duration Interval Analysis

Other than antibiotics for chronic use, we initially classified the remaining antibiotics as appropriate, potentially appropriate, inappropriate, and not associated with any diagnosis based on diagnoses from the same day as the antibiotic prescription. We then examined the effect of expanding the look-back and look-back-and-forward durations between antibiotic prescriptions and diagnosis codes. We started with look back durations because this likely represents better clinical practice (i.e., making a diagnosis followed by an antibiotic prescription). We examined durations as long as 30-days because some analyses use within-month (i.e., up to 30 days) to associate diagnoses and antibiotic prescribing, despite the fact that using these long durations may increase the likelihood of capturing problems unrelated to the antibiotic prescription” [15]. 

We examined look-back periods of −3 days (i.e., a prescription on the day of or 3 days prior to the day of the antibiotic prescription), −7 days, and −30 days. We examined look-back-and-forward periods of ±3 days (i.e., a diagnosis on the day of or the 3 days before or after the day of the antibiotic prescription), ±7 days, and ±30 days.

Owing to large sample sizes, we did not conduct formal statistical significance testing. Rather, we considered absolute differences of 5% or greater to be clinically significant [12,17].

## 3. Results

In the four-year study period from 2016 to 2019, there were 714,057 antibiotic prescriptions ordered for 348,739 unique patients by 2391 clinicians in 467 clinics. Patients had a mean age of 41, were 61% women, 78% white, and 52% had private insurance. Clinicians were 58% women and 59% staff physicians, 19% residents or fellows, and 21% nurse practitioners or physician assistants. 

In all analyses, by design, the proportion of prescriptions classified as chronic infection-related was 16%. Using different look-back and look-back-and-forward durations had modest effects on the distribution of prescriptions across the four remaining appropriateness categories (Table 1). When only considering same-day diagnoses, the proportion of antibiotics classified as antibiotic-appropriate or potentially appropriate was 50%. As the look-back period lengthened, the proportion of antibiotics categorized as appropriate or potentially appropriate increased (59% for a 30-day look-back duration), mostly attributable to more frequent potentially appropriate diagnoses, which increased from 36% for same-day to 43% for −30 days. In contrast, as the look-back periods lengthened, the proportion of antibiotics classified as inappropriate stayed the same, while the proportion of antibiotics associated with no diagnosis decreased. 

Using the longest look-back-and-forward duration (±30 days), potentially appropriate diagnoses accounted for the largest share of prescriptions (44%), followed by inappropriate (20%), antibiotic-appropriate (18%), and not associated with any diagnosis (2%). 

## 4. Discussion

Using different look-back and look-back-and-look-forward periods modestly changed the apparent appropriateness of antibiotic prescribing. Longer durations increased the likelihood of classifying an antibiotic prescription as potentially appropriate, did not change the likelihood of classifying a prescription as inappropriate, and decreased the likelihood of classifying a prescription as not associated with any diagnosis. Diagnoses that sometimes warrant antibiotics (e.g., sinusitis, pharyngitis) are more common than those that always warrant antibiotics (e.g., urinary tract infections, streptococcal pharyngitis) [5,18], so it makes sense that longer durations would capture more of the former, resulting in an increase in the proportion of prescriptions classified as potentially appropriate. The increase in antibiotics classified as potentially appropriate appeared to be due to fewer antibiotics not being associated with any diagnosis.

This analysis has limitations. As with other large data analyses of EHR or claims-based antibiotic prescribing appropriateness, we were dependent on clinician-assigned diagnosis codes and could not assess their validity. When feasible, smaller scale studies by front-line antibiotic stewards should verify clinician diagnoses through chart review. Our findings may not extend perfectly to claims data, which have delays between when the visit is billed, the prescription is filled, and when the filled prescription is billed. Other limitations include the use of a single EHR, conduct in a single US healthcare system with mostly white patients, and data that predate the COVID pandemic. We only examined the overall relationship between the antibiotic prescribing-diagnoses duration and apparent appropriateness as this relationship is a property of the data itself and we have no a priori reason for there to be differential relationships within patient, prescriber, or illness subgroups.

## 5. Conclusions

Ambulatory antibiotic stewardship programs and research projects can select a duration of association that makes sense for their purposes. For example, if the intent is to identify and intervene on inappropriate diagnoses, then the use of a short interval–even same-day–is reasonable, as the proportion of prescriptions classified as inappropriate varies minimally based on the look-back and look-forward periods [11,19,20,21,22]. If the intent is to intervene on potentially appropriate antibiotic prescribing, using longer durations will capture a larger number of these diagnoses [23,24,25]. If seeking to measure and intervene on all ambulatory antibiotic prescribing across all categories of appropriateness, antibiotic stewards and researchers could reasonably select a short look-back duration, like 3 or 7 days. Using a duration of 30 days or adding look-forward codes does not result in major changes in classification decisions. This suggests that researchers who do not favor using such long durations or looking forward for diagnosis codes may avoid doing so without considerably changing estimates of antibiotic appropriateness. However, performance measurement and payment decisions, like those based on the new HEDIS measure “Antibiotic Utilization for Respiratory Conditions (AXR),” are high-stakes applications based on percentile scores among health plans. Interval choices could lead to small absolute differences that are of little clinical importance, but could result in large differences in ranking among prescribers or prescriber groups [7,26].

## Figures and Tables

**Table 1 antibiotics-11-01554-t001:** Different Look-Back and Look-Back-and-Forward Durations and the Apparent Appropriateness of Antibiotic Prescribing (*n* = 714,057 antibiotic prescriptions).

	Same-Day	Look-Back Only			Look Back-and-Forward		
	0 Days	−3 Days	−7 Days	−30 Days	±3 Days	±7 Days	±30 Days
	% *						
Chronic ^†^	16	16	16	16	16	16	16
Appropriate or potentially appropriate	50	53	55	59	54	57	62
*Appropriate*	*14*	*15*	*15*	*16*	*15*	*16*	*18*
*Potentially appropriate*	*36*	*38*	*40*	*43*	*39*	*41*	*44*
Inappropriate	22	24	24	23	24	23	20
No diagnosis	11	7	5	2	6	4	2

* Column percents. Percentages may not add up to 100% because of rounding. We considered differences of 5% or greater to be clinically significant. ^†^ The chronic category always used diagnosis codes and problem list diagnoses in the 6 months prior to the antibiotic prescription.

## Data Availability

Restrictions apply to the availability of these data they are identifiable, protected health information of Northwestern Medicine patients. Making the data externally available would require significant anonymization efforts.

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
