# Peer review of "Look-Back and Look-Forward Durations and the Apparent Appropriateness of Ambulatory Antibiotic Prescribing"

_antibiotics, 2022, doi:10.3390/antibiotics11111554_

Round 1

Reviewer 1 Report

In this retrospective cohort study, the authors used previously reported criteria for outpatient antibiotic prescribing using ICD-10 codes, and assessed the ability of different time frame to capture the appropriateness – same day, -3days, - 7days, -30days, +/-3days, +/-7days, and +/-30days. It seemed the % of chronic antibiotic use did not change at all, inappropriate use did not change greatly, appropriate/potentially appropriate use increased with wider time frame, and no diagnosis decreased significantly. The study was well conducted and presented nicely.

As authors stated, those criteria are solely based on administrative codes and does not verify physician’s diagnosis, which is very often inappropriate. For this reason, those criteria would not be very useful for front-line stewards but would be useful for research purpose or large, multicenter initiatives where chart-review is almost impossible. As a clinician, I agree a few days look back makes sense – as physicians often wait for culture or test results before prescribing antibiotics. I am not sure looking forward makes sense - to call antibiotics given without diagnosis at the time of prescription as appropriate based on the diagnostic code in the future.

Furthermore, I am not convinced to use +/- 30 days time frame, as it is too long for follow-up purpose and it may falsely include another problem from the index diagnosis which antibiotic was prescribed.

Discussion – lines 123-128 is not convincing to me, as I do not think reasonable to use 30 days time frame.

Reviewer 2 Report

From the title, the study has an unclear purpose. This impression is increased by the Data Availability Statement: "The data for this analysis are identifiable, protected health information of Northwestern Medicine and cannot be made available," Moreover, they mentioned to the Institutional Review Board and Informed Consent Statement that "the Northwestern University Institutional Review Board approved this study with a waiver of informed consent for retrospective review of EHR data."

Is this document not registered with a date and number? Were the data available for them, in a suitable proportion for this study, with a lack of concrete data?

Major comments are as follows:

 Introduction

The authors should give a short presentation about the electronic health record (EHR) databases and the most common methods for analysis. They should insist on their utility and relevance in scientific research, with suitable references. Finally, they should specify the novelty of their study compared to similar ones from the literature data.

 Results

This section can begin with a scheme for their study to evidence their reasoning from which they start and what they want to demonstrate.

Why they showed the types of patients and clinicians? In the following data from results, they did not use this information.

Materials and methods

More examples related to their notations and more data from the accessed database are missing for better understanding.

Discussion

The authors should compare their results with those from the literature and provide evidence of the scientific value of their study.

Conclusions are missing.

Round 2

Reviewer 2 Report

       I thank the authors for their efforts to provide the missing data and rectify their manuscript according to previous comments from round 1. 

       However, I have some mentions and suggestions for a more interesting manuscript with the "potential to be widely cited," as the authors felt in their response, page 3/73.

1.   I think it is not a very good idea to begin the consistent conclusions from the "Discussion" section with "In conclusion and practically...". Maybe a short proposition to announce the Conclusions, followed by all original final phrases, should be better. Please, check and correct. 

2.    Please clarify: is this study planned as an article or a brief report? In the front of the manuscript, it is noted Article, and in "Discussion," the authors named it a "brief report" (line 112).

3.     Being only 26 references, I accessed each of them. I read with great interest, especially references 5,12,18, 19, and 26, and I congratulate the co-authors because these articles are very valuable and excellently written; surprisingly, they even include a substantial statistical analysis and "Conclusions" as a separate section.

Moreover, I discovered that EHR database analysis could be extremely attractive, a feeling opposite to the one I had when I first read the present manuscript.

4.    Because I'm still thinking about the giant table, but with very little registered data, which occupies the entire page 3 of 7 and is marked as (Table) in the text (line 80), I have one final suggestion: to develop the presentation of displayed data by correlating with the patient age/sex/race (in various groups), with/without private insurance, the type of professional prescribers, the antibiotics found on prescriptions, maybe with some of the most known codified diseases. 

        Therefore, much smaller but significant groups will also be analyzed, making it possible to do statistical analysis and the corresponding correlations. I have no doubt that the authors, professionals in various medical departments, will reveal and discuss these correlations in the most appropriate way possible.

5.    After all of these were done, I would wholeheartedly believe that this work is up to the standards of a Q1 journal with a high impact factor, such as Antibiotics MDPI.

Author Response

Please see attachment for cover letter and response to Review Report.
